# SpatialHand: Generative Object Manipulation from 3D Prespective

**Zehan Wang**[1*], **Jialei Wang**[1*], **Siyu Chen**[1], **Ziang Zhang**[1], **Luping Liu**[2],
**Xize Cheng**[1], **Kaihang Pan**[1], **Hengshuang Zhao**[2], **Zhou Zhao**[1†]

[1]Zhejiang University; [2]The University of Hong Kong

https://spatialhand.github.io/

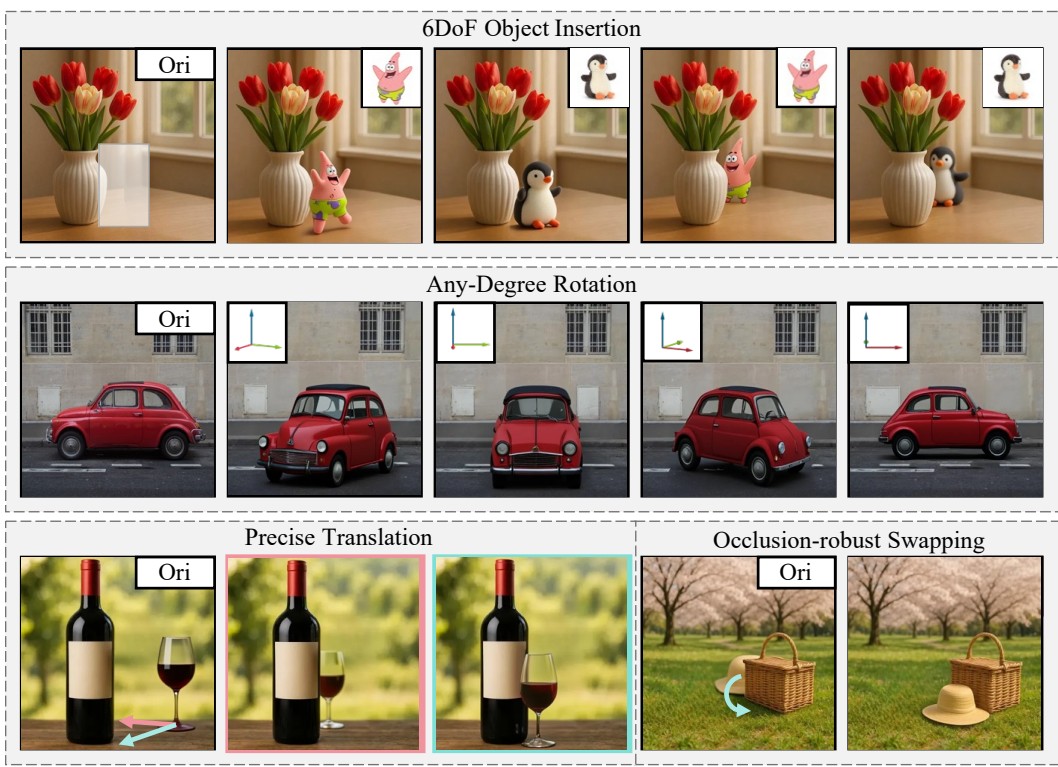

Figure 1: Illustration of how the **"SpatialHand"** can manipulate objects in images from 3D prespective. With the basic ability to insert objects with 6DoF control (top row), allowing for any-degree rotation (middle row), and precise 3D movement (bottom row).

## Abstract

We introduce SpatialHand, a novel framework for generative object insertion with precise 3D control. Current generative object manipulation methods primarily operate within the 2D image plane, but often fail to grasp 3D scene complexities, leading to ambiguities in an object's 3D position, orientation, and occlusion relations. SpatialHand addresses this by conceptualizing object insertion from a true "3D perspective," enabling manipulation with a complete 6 Degrees-of-Freedom (6DoF) controllability. Specifically, our solution naturally and implicitly encodes the 6DoF pose condition by decomposing it into 2D location (via masked image), depth (via composited depth map), and 3D orientation (embedded into latent features). To overcome the scarcity of paired training data, we develop an automated data construction pipeline using synthetic 3D assets, rendering, and subject-driven

---

[*]Equal Contribution.

[†]Corresponding Author.

generation, complemented by visual foundation models for pose estimation. We further design a multi-stage training scheme to progressively drive SpatialHand to robustly follow multiple complex conditions. Extensive experiments reveal our approach's superiority over existing alternatives and its great potential for enabling more versatile and intuitive AR/VR-like object manipulation within images.

# 1 INTRODUCTION

Manipulating an object within the 3D scene of images is a fundamental capability in AR/VR environments and real-world content creation workflows (Biener et al., 2020; Mendes et al., 2019; Yu et al., 2021; Besançon et al., 2021; Monteiro et al., 2021; Gardony et al., 2021; Zhang et al., 2025; Wang et al., 2025b; 2023; Huang et al., 2024; Wang et al., 2025c; Linwei et al., 2026). Despite current generative object insertion and movement methods (Xie et al., 2023; Yang et al., 2023; Huang et al., 2025; Chen et al., 2024a; Podell et al., 2023; Rombach et al., 2022) having achieved significant advancements in object identity preservation and contextual blending, they primarily operate in the 2D plane and lack the understanding or control over the underlying 3D scene layout within the image.

In this work, we revisit generative object manipulation from a 3D perspective. Here, "3D perspective" refers to inserting or moving an object with a specified 6DoF Pose (3D location and 3D orientation), forming correct spatial alignment and occlusion relation with other objects in the scene.

To enable efficient and effective object manipulation in 3D perspective, we propose **SpatialHand**, a practical solution that implicitly encodes an object's 6DoF pose. Although it's challenging for image generation models to directly understand 3D location, we observe that they can effectively follow 2D masks and depth maps. Based on this, we represent the 3D location as a combination of 2D position and depth. By introducing geometry-aware masked images and depth maps, we can precisely control object placement and ensure realistic occlusion relationships. Besides, the 3D orientation is embedded into the latent features as additional guidance for object pose.

Despite our architecture modification enabling the model to natively encode specified 6DoF pose information, the lack of paired training data remains a challenge. To address this, we first leverage advanced 3D generation models (Yang et al., 2024c; Zhao et al., 2025) to create a large collection of high-quality 3D assets. Based on these assets, we introduce two strategies to complementarily produce the target images: 1) Rendering the assets in virtual scenes using a rendering engine (Blender, 2025). 2) Blending the assets with subject-driven image generation models (OpenAI, 2025; Wu et al., 2025). 3) we apply visual foundation models (Liu et al., 2024; Ren et al., 2024; Yang et al., 2024b;a; Wang et al., 2024b; Kirillov et al., 2023; Wang et al., 2025a; 2026) to estimate the object's 6DoF pose from these synthetic images.

Given the multiple types of spatial guidance our model should follow, we design a multi-stage training scheme to progressively teach the model to understand and follow complex spatial cues. We initialize our model with a subject-driven generation model (Wu et al., 2025) to inherit the identity preservation capabilities. Then, we first train on single-object images to drive the model to generate novel views of objects with a specified 3D orientation. Afterwards, we use multi-object complex scenes to learn further controllable 3D positioning and achieve realistic occlusion relationships.

We evaluate our method both quantitatively and qualitatively across various tasks and scenes. The results demonstrate a new level of controllability in object insertion from the 3D perspective, enabling intuitive AR/VR-like object manipulation and several novel 3D-aware content creation applications.

In summary, our contributions are four-fold:

- We highlight the "3D perspective" of the generative object manipulation task, aligning with how humans manipulate objects in the AR/VR environment. Our approach enables precise control over object 6DoF pose and eliminates the ambiguity in 2D image inpainting.

- We decompose the 6DoF pose into 2D location, depth, and 3D orientation, allowing our model to naturally and implicitly encode spatial condition information.

- We develop a stable and automated training data curation pipeline using high-quality synthetic 3D assets, rendering engine, and subject-driven image generation models.

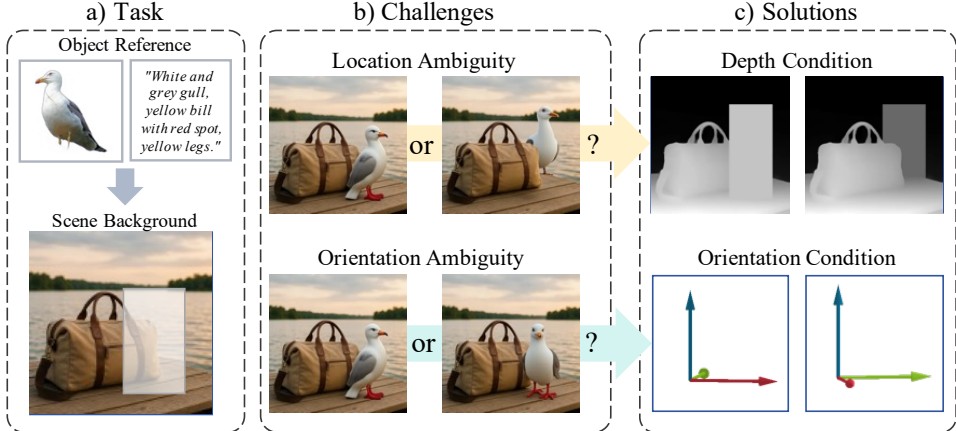

Figure 2: **Motivation of SpatialHand.** 2D inpainting for object insertion/movement suffers from location ambiguity (in front of or behind existing elements?) and orientation ambiguity (facing right or left?). SpatialHand resolves these by adding extra depth and orientation conditions for spatially controlled object manipulation.

- We employ a multi-stage, decoupled training approach that enables the model to progressively adapt to complex spatial conditions, achieving robust spatial control.

## 2 RELATED WORK

### 2.1 GENERATIVE OBJECT MANIPULATION

With the advancement of generative models (Rombach et al., 2022; Podell et al., 2023; Peebles & Xie, 2023; Labs, 2024), many methods now use inpainting to place objects into scene images, guided by visual or textual conditions. Text-guided inpainting methods (Zhang et al., 2023a; Avrahami et al., 2022; Yu et al., 2023) can easily generate objects described in text within the inpainting mask of existing images. Paint-by-Example (Yang et al., 2023) and AnyDoor (Chen et al., 2024a) enable visual reference objects by using visual features as additional conditions. UniReal (Chen et al., 2024b) and ObjectMover (Yu et al., 2025) take it a step further by leveraging prior knowledge of video generation models, using data from videos or game engines to achieve more realistic and plausible object replacement.

Although these methods perform well in preserving object identity and environmental interactions, their reliance on simple 2D inpainting masks leads to ambiguity in the results. As shown in Fig. 2, they cannot determine whether an inserted object should appear in front of or behind existing objects, or whether it should face left or right. To address this, our work considers the object manipulation task from the 3D perspective, enabling precise object insertion & movement with a specified 3D location and pose.

### 2.2 3D-AWARE IMAGE EDITING

Recently, several works (Michel et al., 2023; Wu et al., 2024; Yenphraphai et al., 2024; Wang et al., 2024a; Pandey et al., 2024) have tried to edit and manipulate the content in 2D images from the 3D space, like location translation and object rotation. Object-3DIT (Michel et al., 2023) collects a synthetic dataset and learns 3D-aware image editing guided by language instructions. Diffusion Handles (Pandey et al., 2024) and Diff3DEdit (Wang et al., 2024a) convert 2D images into 3D point clouds to manipulate objects directly in 3D space. Image Sculpting (Yenphraphai et al., 2024), on the other hand, models the object's 3D mesh and performs edits on the mesh itself. These point cloud- or mesh-based methods use complicated techniques to maintain consistency in the resulting images, such as feature injection, DreamBooth tuning (Ruiz et al., 2023), ControlNet (Zhang et al., 2023b), and noise inversion (Song et al., 2020).

Although existing methods offer explicit 3D control, they introduce significant complexity and latency, and often fail to capture the full geometric structure of the 2D image. For example, point

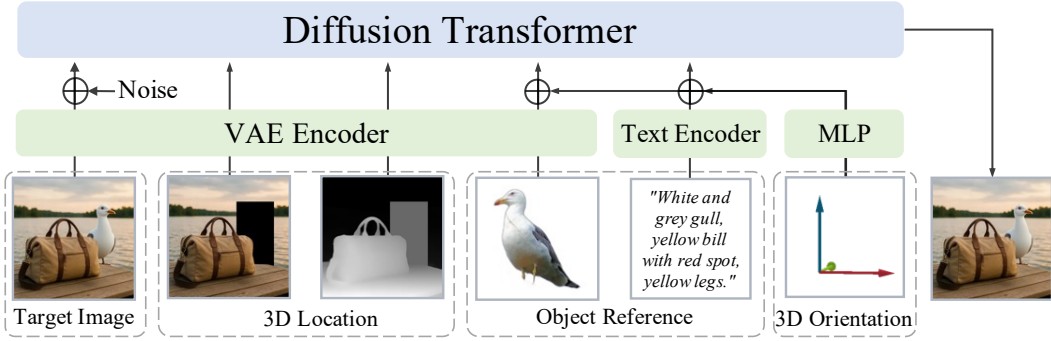

Figure 3: **Overall pipeline of SpatialHand.** We focus on object insertion as our primary task due to its flexibility. SpatialHand decomposes the 6DoF object pose into 3D location (2D mask and depth map) and 3D orientation. These spatial conditions, along with free-view object reference images and text captions, are incorporated into the diffusion transformer's input tokens.

clouds cannot represent the back side of objects, limiting large rotations, while synthetic meshes struggle with reshaping occlusion relationships. In contrast, our method leverages implicit 3D guidance and a one-stage image generation model, significantly simplifying the whole process while supporting more diverse and comprehensive 3D editing capabilities.

## 3 METHOD

Our main goal is to place an object into an image with a specific 6DoF pose (3D location and orientation), and form realistic occlusion relationship. To this end, we modify the standard diffusion transformer model to enable decomposed spatial condition input (Sec. 3.1), curate large-scale training data pairs (Sec. 3.2) and introduce a progressive training scheme (Sec. 3.3) to effectively teach the model to follow spatial guidance.

### 3.1 MODEL DESIGN

**Preliminary**  Our framework is built upon the advanced open-sourced text-to-image model, FLUX-Dev.1 (Labs, 2024), which employs the MM-DiT structure. The original MM-DiT block takes the concatenation of noisy image tokens $\mathbf{X} \in \mathbb{R}^{N \times D}$ and text condition tokens $\mathbf{C}_\mathrm{T} \in \mathbb{R}^{L \times D}$ as input, where $N$, $L$ are the length of image and text tokens, and $D$ is the latent dimension. The combined token sequence is projected into Query $\mathbf{Q}$, Key $\mathbf{K}$, and Value $\mathbf{V}$ representations, and is globally interacted with the multi-modal attention mechanism:

$$\mathrm{Attn}([\mathbf{X}, \mathbf{C}_\mathrm{T}]) = \mathrm{Softmax}(\frac{\mathbf{Q}\mathbf{K}^\mathrm{T}}{\sqrt{\mathrm{d}}})\mathbf{V} \qquad (1)$$

To repurpose the standard text-to-image generation model for object insertion, we first extend the input sequence with background and object reference conditions. Specifically, we embed the masked scene image $\mathbf{C}_\mathrm{mask}$ and the reference object $\mathbf{C}_\mathrm{obj}$ (provided as image and caption) into tokens with the pretrained VAE and text encoder, and concatenate them with the noisy image tokens as additional conditional inputs. The results sequence can be formulate as $[\mathbf{X}, \mathbf{C}_\mathrm{mask}, \mathbf{C}_\mathrm{obj}]$.

**3D Location Condition**  Using only the masked scene and reference object, as previous methods, provides limited 2D positional control. A simple 2D mask cannot determine an object's precise location in actual 3D space, leading to both location and orientation ambiguities, as shown in Fig. 2.

To resolve these ambiguities and enable precise 3D positioning, we further introduce an additional composited depth map to explicitly define the desired insertion depth. We first predict the scene's depth map using Depth Anything (Yang et al., 2024b), then modify the depth values within the masked region to match the target placement. In real practice, this target depth can be user-friendly and intuitively specified by simply clicking on the desired ground position.

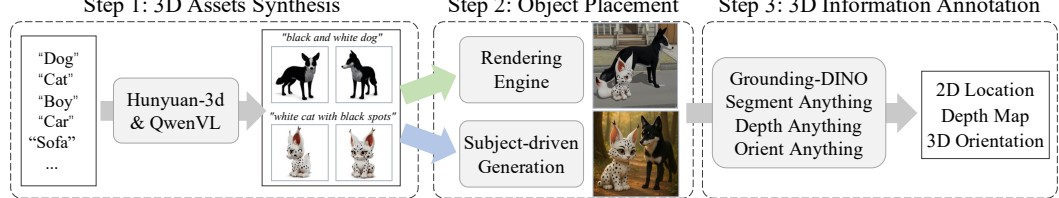

Figure 4: **Pipeline of training data curation.** We start with high-quality synthetic 3D assets. Using a rendering engine and subject-driven generation, we simulate how humans place objects in 3D space. Then, we employ a series of visual foundation models to estimate the 3D information within images.

To preserve realistic occlusions and foreground consistency, we further propose geometry-aware composition. By comparing the scene's depth map and the intended placement depth, we identify masked foreground objects that should remain visible. These objects, in front of the insertion place, are preserved in both the masked scene and depth map, ensuring correct occlusion (i.e., foreground objects stay in front of the inserted object).

In summary, our method extracts the scene's depth map, marks the depth value of the desired location on it. Besides, we further use geometry-aware composition to refine the occlusion relationship and produce the geometry-aware masked image $\tilde{\mathbf{C}}_{\text{mask}}$ and depth map $\tilde{\mathbf{C}}_{\text{depth}}$. The input token sequence is further extend to $[\mathbf{X}, \tilde{\mathbf{C}}_{\text{mask}}, \tilde{\mathbf{C}}_{\text{depth}}, \mathbf{C}_{\text{obj}}]$.

**3D Orientation Condition**     After determining the 3D location, our method additionally enables specification of the object's 3D orientation to achieve full 6DoF controllability. Previous novel view synthesis methods (Liu et al., 2023; Wu et al., 2024) primarily operate through relative rotations from reference views, limiting their effectiveness in textual-only scenarios. In contrast, we focus on modeling semantic object orientation that aligns with an object's canonical front-facing direction. Our method incorporates absolute target orientations as direct conditions rather than relying on relative rotation.

We describe the 3D orientation of an object with three parameters: azimuth $\varphi$, elevation $\theta$, and in-plane rotation $\delta$, following Orient Anything (Wang et al., 2024b; 2026). To introduce the orientation parameters as a condition for the diffusion transformer, we apply a zero-initialized MLP projector $P(\cdot)$ to map them into the latent dimension and add them into the object reference conditions. The final input token sequence can be formulated as $[\mathbf{X}, \tilde{\mathbf{C}}_{\text{mask}}, \tilde{\mathbf{C}}_{\text{depth}}, \mathbf{C}_{\text{obj}} + P([\varphi, \theta, \delta])]$.

## 3.2 Training Data Curation

For placing object with specific 6DoF pose, ideal training data pairs should cover: 1) *Object Condition:* textual caption and visual image from difference view for the reference object, 2) *Target Image:* target scenes with placed object, and *3) 6DoF Guidance:* object location, depth and orientation annotations. The lack of paired data is another obstacle for 3D-aware object replacement. To overcome this, we develop an automated data curation pipeline that simulates real-world object placements through three steps:

**Step 1: 3D Object Assets Synthesis**     In the real world, the first step of placing objects is obtaining the actual 3D-form object. Inspired by recent advances in 3D generation models (Yang et al., 2024c; Zhao et al., 2025; Xiang et al., 2024), we synthesize 3D object assets from category names for comprehensive data coverage. With the high-quality 3D mesh, the multi-view renderings can serve as visual object conditions, and associated captions are native textual object conditions. Specifically, we leverage Hunyuan-3D 2.0 (Zhao et al., 2025) to generate 43k 3D assets from 10k common object category names, render 20 random views for each object, and employ Qwen-2.5-VL (Bai et al., 2025) to produce corresponding captions.

**Step 2: Object Placement Simulation**     To mimic the process of placing 3D objects in the real world, we design two complementary object placement methods: 1) *Simulated Placement:* Using

the Blender simulation platform (Blender, 2025), we randomly arrange and render 3D assets to generate diverse 6DoF poses and occlusion patterns across scenes. This approach ensures perfect object identity preservation but sacrifices scene diversity and complexity. 2) *Generative Placement:* Leveraging subject-driven image generation models, UNO (Wu et al., 2025) and ChatGPT-4o (OpenAI, 2025), we blend multiple objects into realistic scenes with natural arrangements. This method enhances contextual diversity and plausibility, albeit with slightly weaker identity consistency. By combining the two complementary strategies, we balance geometric and identity precision (Blender) with rich environmental variation (generative models), providing scalable, diverse, and reliable target images.

**Step 3: 3D Information Estimation**   Given the visual and textual reference object along with the target image containing the reference object, the final step involves estimating the object's spatial pose within the target image as 6DoF guidance. First, we use Grounding-DINO (Liu et al., 2024) and Segment Anything (Kirillov et al., 2023) to detect the 2D position (bounding box and segmentation mask). Second, the target image's depth map is predicted using Depth Anything (Yang et al., 2024b), and the placed objects' average depths are computed within its segmented region. Finally, we infer the 3D orientation of the reference object using Orient Anything (Wang et al., 2024b; 2026).

**Statistic**   We collect 43k diverse 3D assets spanning a wide range of object categories. Using simulated placement and generative placement techniques, we generate 100k and 450k target images, respectively. To ensure data quality, we implement a rigorous filtering process that removes: (1) samples with low DINO similarity scores to reference objects and (2) cases with low bounding box or orientation confidence. This quality control process results in 370k high-fidelity object-image-pose training pairs.

## 3.3 TRAINING SCHEMES

Achieving generative 3D-aware object placement and manipulation necessitates that the model adhere to several conditions: object identity, 2D location, depth map, and 3D orientation. To facilitate robust learning amidst these intricate constraints, we introduce a progressive training scheme.

**Stage 0: Identity-Preservation Pre-training**   Object identity preservation is the most fundamental ability of our tasks. Therefore, we initialize our model with the pre-trained FLUX-1 dev (Labs, 2024) model and the LoRA adapter (Wu et al., 2025) trained for subject-driven generation tasks, ensuring the model inherently possesses object identity preservation capabilities from the beginning of training.

**Stage 1: Novel View Synthesis Fine-Tuning**   Then we focus on training the model to comprehend object orientations and generate novel views with specific 3D orientations. To this end, we randomly select two renderings of the same 3D object as object reference and target images, and incorporate estimated orientation predictions from Orient Anything as guidance for training.

**Stage 2: 3D-aware Insertion Fine-Tuning**   Finally, we refine the model's ability to realistically insert objects into scenes with specified poses. Using the dataset introduced in Section 3.2, we train the model to accurately position objects while maintaining background.

**Implementation Details**   We initialize our model using the pre-trained FLUX-1 dev (Labs, 2024) framework, along with the LoRA adapter provided in UNO (Wu et al., 2025). For stages 1 and 2 in Section 3.3, we employ the AdamW (Loshchilov & Hutter, 2017) optimizer with a learning rate of 1e-4 and batch size of 8 , using a cosine scheduler. We utilize LoRA (Hu et al., 2022) with the rank of 512 for fine-tuning, and the model is trained for 60k steps in Stage 1 and 20k steps in Stage 2. All the experiments are conducted on 8×A100 GPUs.

## 4 EXPERIMENTS

### 4.1 3D-AWARE OBJECT INSERTION

**Alternative Baseline**   Considering there is no existing object insertion method that supports pose condition, we select the advancing image editing model, GPT-4o (OpenAI, 2025), Gemini-2.0-

| | Visual+Textual Condition | | | | | Textual-only Condition | | | | |
| --- | --- | --- | --- | --- | --- | --- | --- | --- | --- | --- |
| | Objective | | | Subjective | | Objective | | | Subjective | |
| | DINO↑ | AbsRel↓ | Acc@30°↑ | Fidelity↑ | Adherence↑ | CLIP↑ | AbsRel↓ | Acc@30°↑ | Fidelity↑ | Adherence↑ |
| Gemini-2.0-Flash | 81.2 | 33.2 | 16.0 | 3.51 | 2.10 | 65.7 | 32.3 | 18.6 | 4.10 | 1.87 |
| GPT-4o | 80.5 | 38.6 | 20.2 | 4.13 | 2.67 | 66.1 | 47.5 | 19.1 | **4.31** | 2.13 |
| Nano Banana | 80.9 | 32.1 | 17.5 | 3.63 | 2.25 | 66.5 | 32.5 | 19.5 | 4.25 | 1.97 |
| SpatialHand | **81.7** | **19.8** | **52.0** | **4.30** | **4.27** | **72.5** | **18.7** | **47.0** | 4.25 | **4.53** |

Table 1: Quantitative comparison results on 3D-aware object insertion. Object similarity in the generated image to the reference is measured using DINO↑ and CLIP↑. AbsRel↓ evaluates the accuracy of the inserted object's depth position, and Acc@30°↑ measures its orientation accuracy.

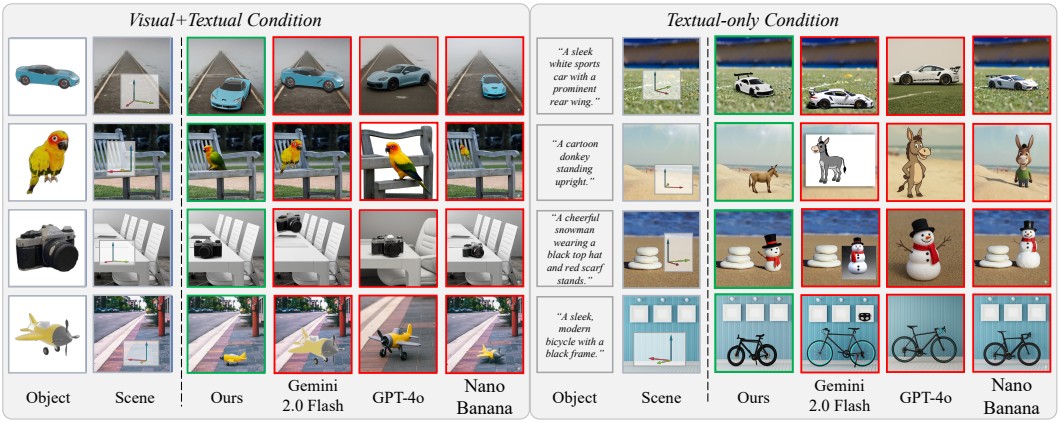

Figure 5: Qualitative comparison on 3D-aware object insertion. Red arrow indicates the desired orientation, and green arrow denotes the left side of the corresponding pose. Zoom in for best view.

Flash (Google, 2025b) and Nano Banana (Gemini-2.5-Flash Image) (Google, 2025a), as our baselines. To incorporate pose guidance, we input the background scene, composited depth map, and reference object into the unified image generation models, along with a detailed instruction specifying the desired 3D location and orientation. Detailed examples are provided in the Appendix.

**Benchmark** Specifically, we manually select 20 images (including both real and synthetic samples) as scene images, and choose 20 high-quality 3D objects from Objaverse (Deitke et al., 2023) as reference objects. We use the renderings and captions from Qwen2.5VL of 3D assets as visual and textual object conditions, respectively. For each scene image, we manually annotate two suitable poses as insertion targets. Our benchmark contains 800 test samples for different kinds of object conditions input (visual+textual and textual-only), amounting to 1,600 test samples in total.

**Evaluation Metric** For identity preservation, we compute the DINO (Oquab et al., 2023) feature similarity between the result image and the rendering under the target pose of the reference object, and the CLIP (Radford et al., 2021) feature similarity between the result image and the textual conditions. For depth alignment, we compute the absolute relative depth error (AbsRel) between the target depth and the inserted object's depth. For orientation alignment, we calculate the accuracy within a 30° angular error tolerance (Acc@30°) between the target and the actual orientation of the inserted object. We also conduct human subjective study, where annotators rate each result image from 1 to 5 based on two criteria: "Object Fidelity" and "Pose Adherence".

**Main Results** Comparative results are provided in Tab. 1 and Fig. 5. First, SpatialHand shows state-of-the-art object ID consistency in both visual+textual and textual-only inputs. More importantly, objects inserted by our method adhere better to the specific pose regarding depth and orientation (lower AbsRel and higher Acc@30°). Even the most advanced instruction-based image editing model, GPT-4o, struggles to understand and maintain the desired spatial state from textual instruction, highlighting the necessity of our carefully designed 3D location and orientation conditions.

| | Rotation | | | Translation | | | | Occlusion | | |
|---|---|---|---|---|---|---|---|---|---|---|
| | *Objective* | *Subjective* | | *Objective* | | *Subjective* | | *Objective* | *Subjective* | |
| | Acc@30°↑ | Fidelity↑ | Adherence↑ | mIoU↑ | AbsRel↓ | Fidelity↑ | Adherence↑ | VLM-Acc↑ | Fidelity↑ | Adherence↑ |
| Object3DiT | 31.4 | 2.78 | 3.22 | 0.45 | 35.4 | 2.36 | 3.56 | 55.2 | 1.67 | 2.31 |
| Diffusion Handles | 20.7 | 3.01 | 3.05 | 0.28 | 20.7 | 2.76 | 3.34 | 49.7 | 1.78 | 2.05 |
| Gemini-2.0-Flash | 18.3 | 3.43 | 2.56 | 0.22 | 27.3 | 3.27 | 2.65 | 49.2 | 3.13 | 2.67 |
| GPT-4o | 19.5 | 3.87 | 2.67 | 0.24 | 38.2 | 3.74 | 2.37 | 59.5 | **3.68** | 3.16 |
| Nana Banana | 20.5 | 3.53 | 2.73 | 0.25 | 26.3 | 3.53 | 2.72 | 50.1 | 3.21 | 2.78 |
| SpatialHand | **47.8** | **3.89** | **4.17** | **0.72** | **17.9** | **4.12** | **4.56** | **82.6** | 3.57 | **4.53** |

Table 2: Quantitative comparison on 3D-aware object movement.

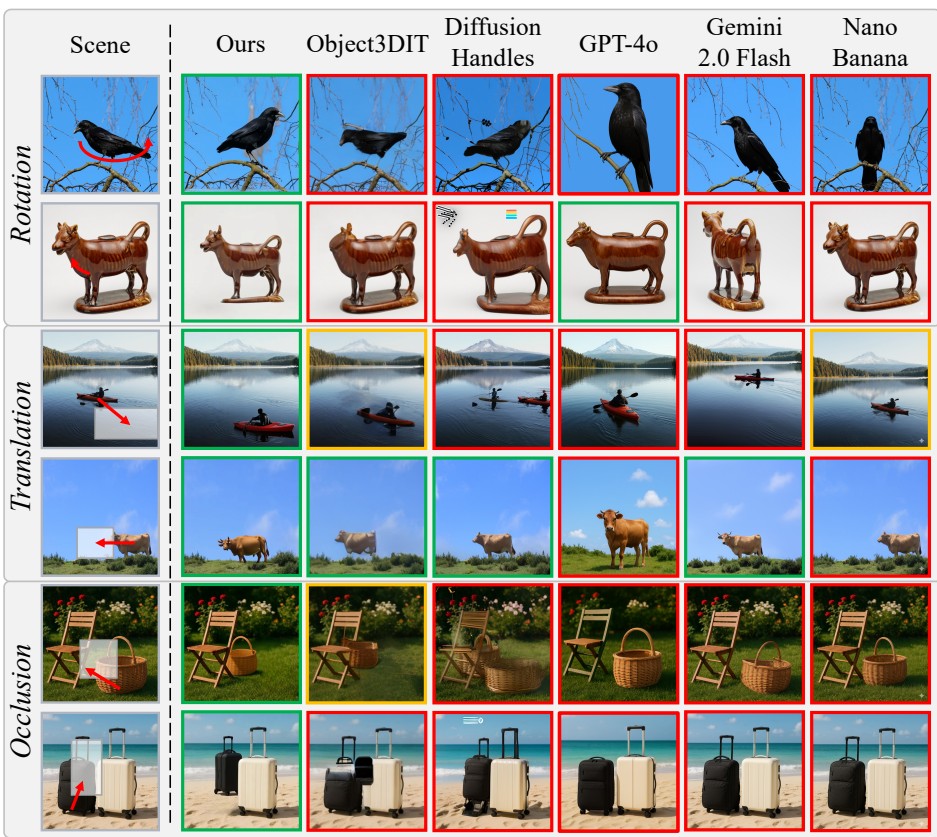

Figure 6: Qualitative comparison on 3D-aware object movement.

## 4.2 3D-AWARE OBJECT MOVEMENT

**Alternative Baseline**  For 3D-aware object movement task, we compare several 3D-aware image editing methods, including Object3DiT (Michel et al., 2023) and Diffusion Handles (Pandey et al., 2024), which support 3D object rotation and translation in existing images. Additionally, we also include advanced unified instruction-based image editing models, Gemini-2.0-Flash (Google, 2025b), GPT-4o (OpenAI, 2025) and Nano Banana (Gemini-2.5-Flash Image) (Google, 2025a), as baselines. Our spatialHand method achieves 3D movement by following two processes: removing the object, then inserting it back with the target pose.

**Benchmark**  We divide 3D-aware object movement into three sub-tasks: 1) horizontal rotation, 2) 3D translation, and 3) handling occlusion (moving objects between foreground and occluded positions). For each sub-task, we collect 50 images and annotate 2 manipulations per image, resulting in 100 samples per sub-task and 300 samples in total.

**Evaluation Metric**  We measure accuracy within 30° angular error (Acc@30°) with Orient Anything, and translation adherence, average box IoU (mIoU) and absolute relative depth error (AbsRel), using DINO-grounding and Depth Anything. For occlusion, advanced VLM, Qwen2.5VL (Bai et al.,

| Row | Geometry | Training stage | Input condition | DINO↑ | AbsRel↓ | Acc@30°↑ |
|-----|----------|----------------|-----------------|-------|---------|----------|
| 1 | Yes | Stage 1&2 | Visual+Textual | **81.7** | 19.8 | **52.0** |
| 2 | Yes | Stage 1&2 | Visual Only | 78.8 | 18.5 | 46.8 |
| 3 | No | Stage 1&2 | Visual+Textual | 80.2 | 25.5 | 48.8 |
| 4 | Yes | Stage 2 | Visual+Textual | 79.8 | **18.2** | 28.7 |

Table 3: Ablation study on 3D-aware object insertion task. "Geometry" indicates whether geometry-aware composition is used for 3D location condition (Sec. 3.1). "Training Stage" refers to different training strategies (Sec. 3.3). "Input Condition" represents different forms of object reference.

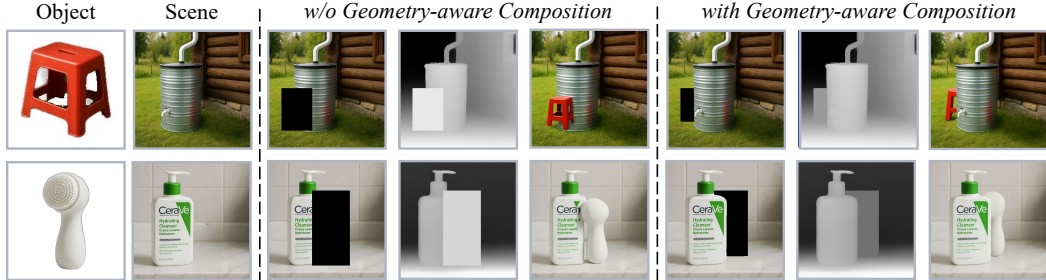

Figure 7: Effect of geometry-aware composition. We sequentially present the masked scene, depth map, and synthesized images, with and without applying geometry-aware composition.

2025), is tasked to check occlusion correctness (VLM-Acc). Human studies about "Object Fidelity" and "Pose Adherence" are also employed.

**Main Results**   Tab. 2 presents the quantitative results for different 3D-aware object movement tasks. Our model exhibits superior advantages over each sub-task. On representative subjective metrics, our method show significantly better object fidelity and spatial state adherence compared to both specialized 3D-aware image editing methods and general image editing methods. Fig. 6 further provides qualitative comparisons of different methods on various subtasks. Methods that explicitly operate on point clouds, like Diffusion Handlers, struggle with large rotations and occlusions due to missing back structure in monocular point clouds. Furthermore, methods relying on textual instructions often fail to describe the complex spatial editing intentions. In contrast, our method implicitly encodes the desired spatial state, enabling diverse 3D-aware editing while avoiding ambiguity.

## 4.3 ABLATION STUDY

**Impact of Textual Condition**   First, we examine the effect of the textual object condition. Comparing rows 1 and 2 in Tab. 3 shows that extra caption information significantly helps maintain object identity during insertion. Considering that VLMs perform well and are widely used for captioning tasks, using VLM captions for visual-only scenarios is a practical choice for real-world applications.

**Geometry-aware Composition**   Fig. 7 visualizes the geometry-aware composition process described in Sec. 3.1. It shows that the geometry-aware masked image and depth map better capture occlusion between inserted and existing objects, and also preserve foreground object identity. Moreover, the higher depth error of row 3 compared to row 1 in Tab. 3 quantitatively demonstrates that geometry-aware composition is effective for controlling object position.

**Progressive Training Scheme**   Row 4 in Tab. 3 shows the result of removing stage 1 from our progressive training (Sec. 3.3). We observe that skipping the fine-tuning for novel view synthesis markedly harms the model's orientation-following capability, as indicated by the lower orientation accuracy, Acc@30°. This confirms our progressive training scheme's effectiveness in helping the model understand object orientation and handle complex conditions.

## 5 CONCLUSION

We introduce SpatialHand to virtually insert or move objects in existing images, operating from a "3D perspective." It uniquely encodes the 6DoF object pose as 2D location, depth, and 3D orientation cues. To overcome data scarcity and ensure robust adherence to complex spatial conditions,

we develop an automated data curation pipeline and a progressive training scheme. Extensive experiments demonstrate SpatialHand's superiority over existing approaches, showcasing its strong potential to advance more intuitive and versatile VR/AR-like object manipulation within images.

## ACKNOWLEDGEMENTS

This work was supported by National Natural Science Foundation of China under Grant (No.62572423, No.624B2128, No.62422606), Hong Kong Research Grant Council General Research Fund (No.17213925) and State Grid Zhejiang Electric Power Cooperation Technology Project (No.B311DS240012)

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

# A   MORE VISUALIZATION COMPARISON

Figure 8: More qualitative comparison on 3D-aware object insertion

More visualization results on 3D-aware object insertion are shown in Figure 8. It shows that our method achieves excellent performance in both 3D position and orientation control during object insertion, while maintaining the consistency of the inserted objects. In contrast, GPT-4o and Gemini-2.0-Flash perform poorly in terms of orientation control and background preservation.

Figure 9 shows more visualization results about object movement. Our method can perform better in rotation, translation, and occlusion handling, while the other four models show lower success rates in operations. Diffusion Handles and GPT-4o perform poorly in background preservation. Gemini-2.0 demonstrates a lack of understanding of bounding boxes.

# B   PROMPT FOR GPT-4O, GEMINI-2.0-FLASH AND NANO-BANANA

For a fair evaluation of GPT-4o and Gemini-2.0-Flash's performance, we use the following prompt for object insertion: "Insert the item in the second picture into the first picture(background), and depth map is provided as the third picture, the boxed area's depth is the average depth of the item should be placed, the caption of the item is {caption}, the item should orient to camera with azimuth {orientation[0]}, polar {orientation[1]}, rotation {orientation[2]},{'common knowledge'}" where 'common knowledge' includes the coordinate system and definitions of the three rotation angles. This prompt contains all the information that we input into our model.

For GPT-4o, we utilize the mask functionality in the API to constrain the regions where objects can be modified during insertion. However, from the results, GPT-4o does not strictly follow this mask, exhibiting cases where objects extend beyond the bounding box and background changes occur.

For object movement, we draw the bounding box of the object in the background to indicate the target position with the prompt: "Move the item in the picture to the new position, which is marked by the red box. Depth map is provided as the third picture, the boxed area's depth is the average depth of the item should be placed. The caption of the item is {caption}, the item should orient to camera with azimuth {orientation[0]}, polar {orientation[1]}, rotation {orientation[2]} ,{'common knowledge'}"

# C   QUANTITATIVE ABLATION STUDY

For the ablation study of Geometry-aware Composition, we have shown the results in Figure 7. Figure 10 visualizes results for the ablation study of the Impact of Textual Condition and Progressive Training Scheme.

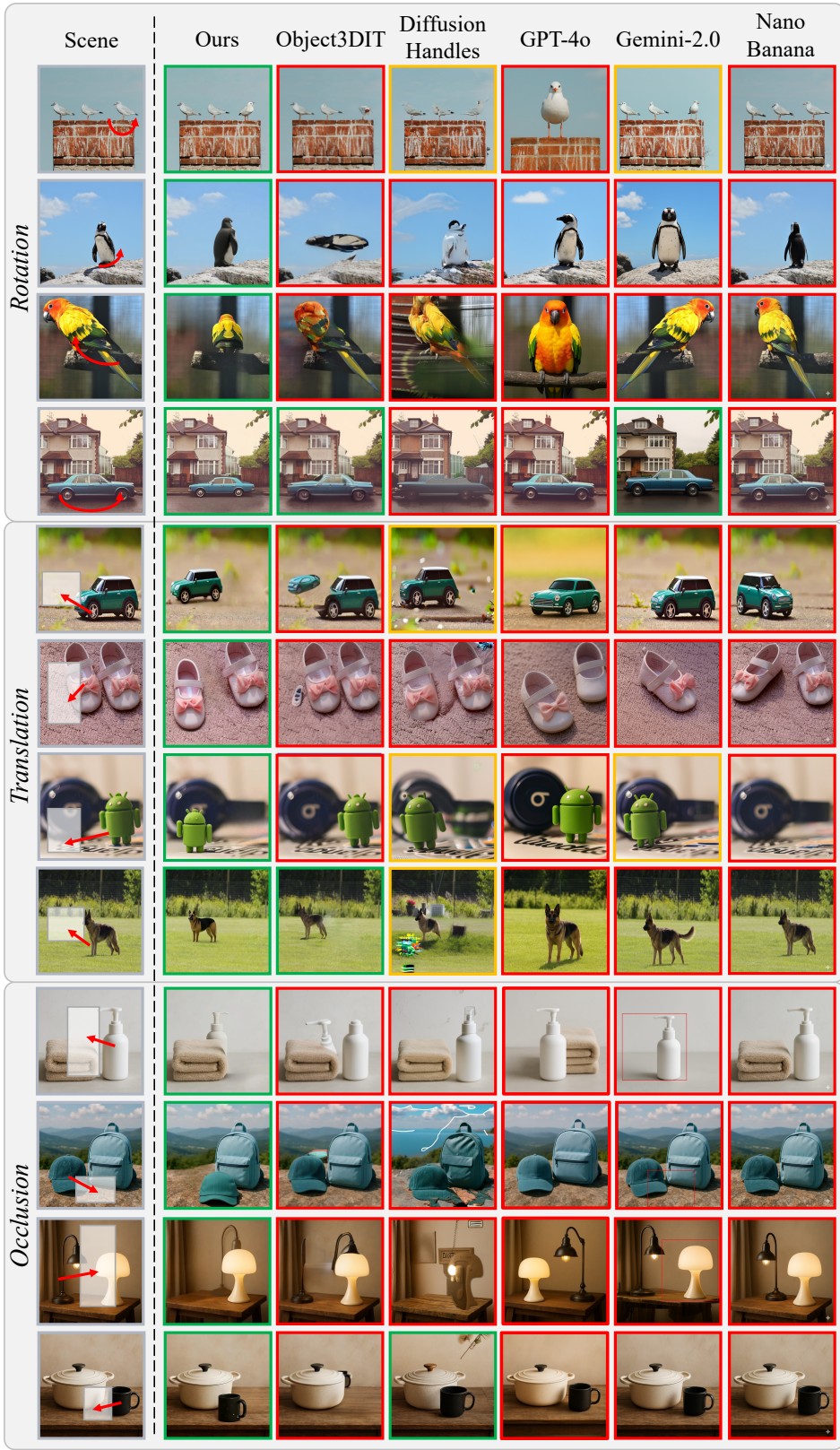

Figure 9: More qualitative comparison on 3D-aware object insertion

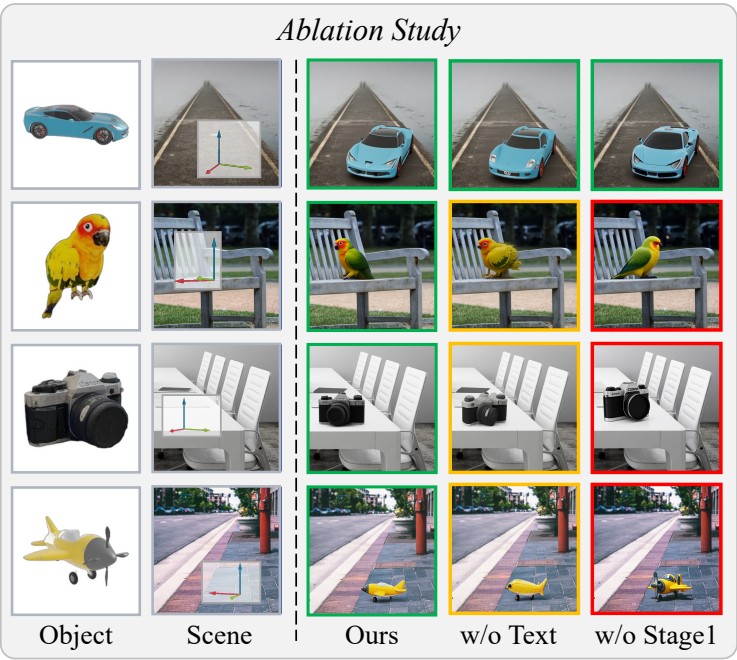

Figure 10: More qualitative comparison on 3D-aware object insertion

The results show that Textual Condition plays a crucial role in maintaining object consistency when changing object orientation. Without the Textual Condition, the features of the inserted object cannot be fully preserved. Additionally, single-stage training shows poor performance in orientation control, which demonstrates the effectiveness of our Progressive Training Scheme.

## D  MORE VISUALIZATION UNDER CHALLENGING SCENARIOS

We provide more visualization results for object movement involving human and artistic subjects in Figure 11 and object insertion in scenes with more diverse lighting conditions in Figure 12. The results show that even though our method was not specifically trained for these situations, it still demonstrates generalization ability on these challenging cases.

## E  USAGE OF LARGE LANGUAGE MODELS

We utilized Large Language Models (LLMs) for sentence-level refinement during the drafting of this manuscript. Our experiments featured several key baselines that are Multimodal Large Language Models (MLLMs), including GPT-4o, Gemini-2.0-flash, and Nano-Banana. Section B thoroughly explains how we prompted and used these MLLMs.

## F  LIMITATION AND FUTURE WORK

Under the current paradigm, an object's orientation and its 2D bounding box are somewhat interdependent. The box's aspect ratio should generally match the object's expected shape under the specific orientation. For instance, a car's side view requires a wider inpainting box than its front view. A significant conflict between these can make it harder for the model to apply the orientation correctly. Making location and orientation conditions more independent is a meaningful future direction.

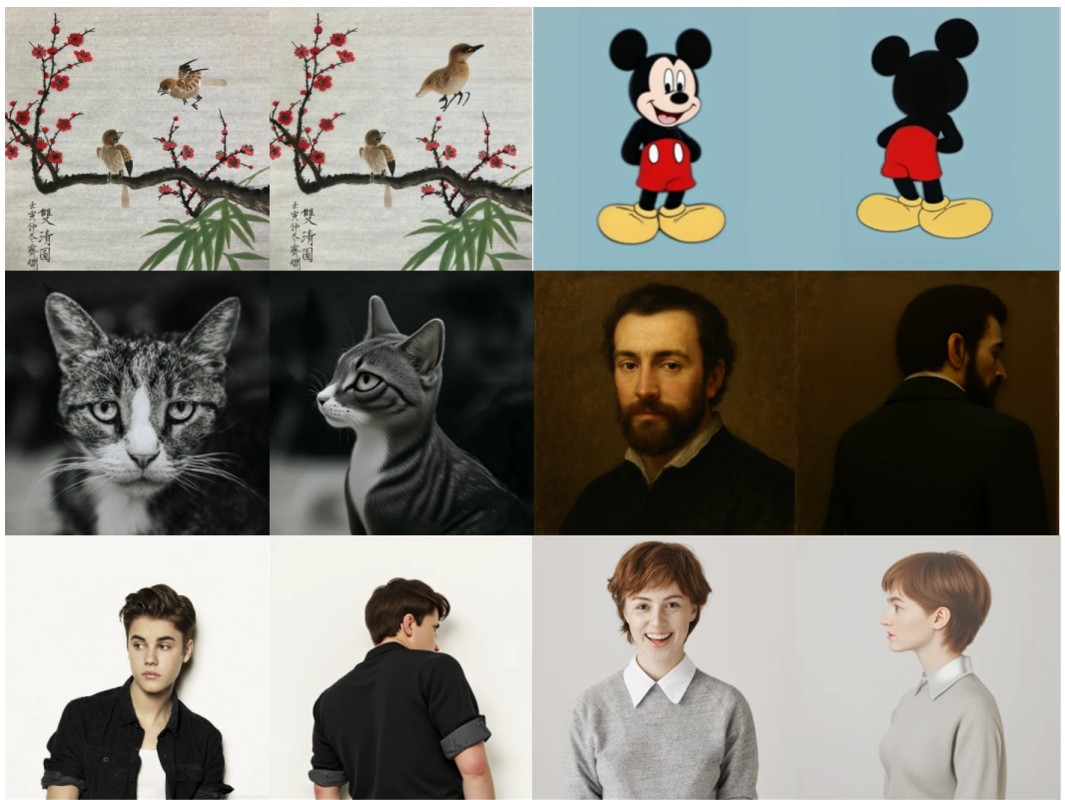

Figure 11: Qualitative results of moving human and artistic subjects.

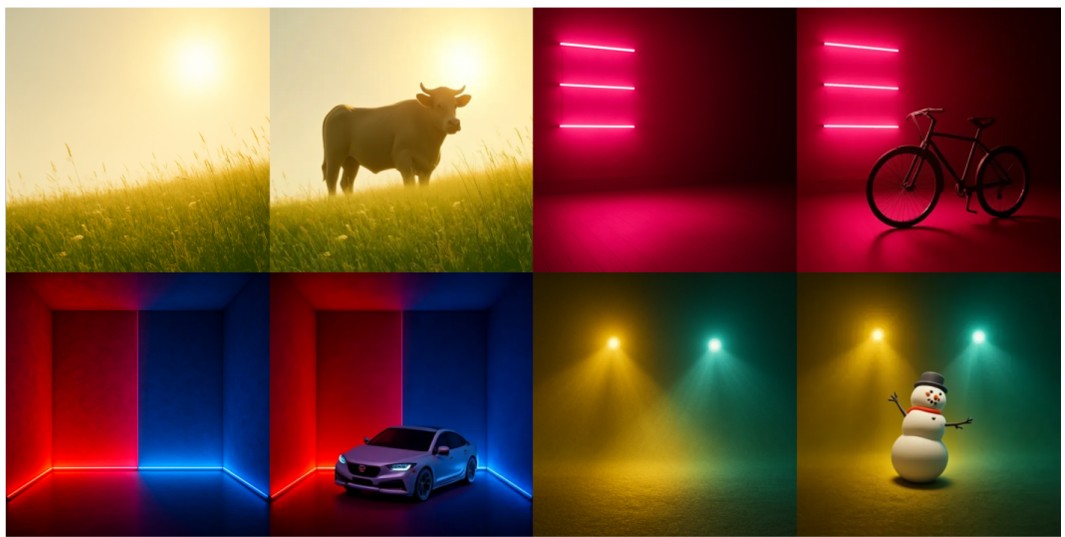

Figure 12: Qualitative results under different and challenging lighting conditions.

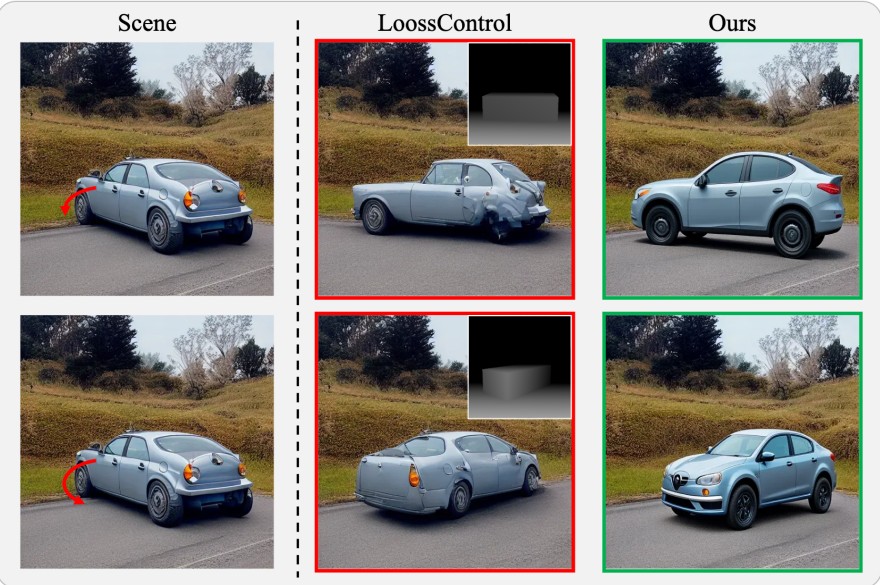

Figure 13: Qualitative results against depth-based methods. Visual comparison demonstrating the inherent ambiguity of the Box Depth Map. Our method accurately executes the 3D rotation while preserving the object's structure.

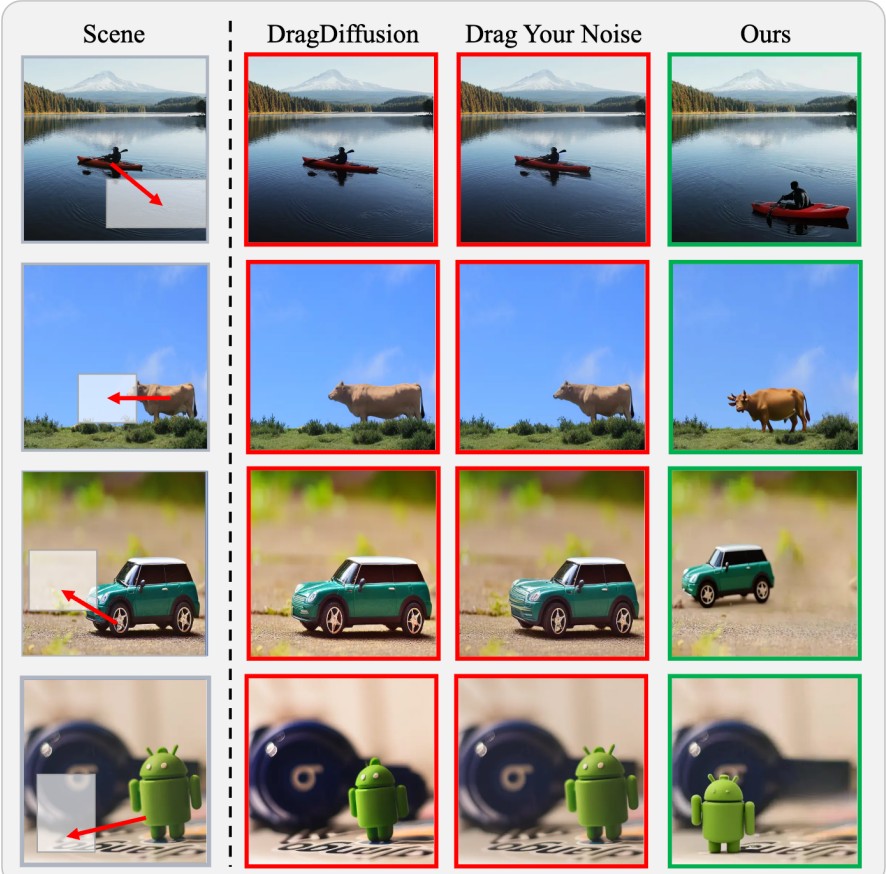

Figure 14: Qualitative results against drag-based methods. Visual comparison showing that drag-based methods tend to cause object deformation when moving objects. Our method successfully performs realistic object translation while maintaining structural integrity.

