# OpenReview forum: "SpatialHand: Generative Object Manipulation from 3D Prespective"
_ICLR.cc/2026/Conference — ICLR 2026 Poster_

### Official Review · Reviewer_XPg4 · 2025-10-23

**Soundness:** 3
**Presentation:** 3
**Contribution:** 3
**Rating:** 8
**Confidence:** 3

**Summary:**

The paper proposes a method for image manipulation to control the 6DoF of objects appearing in it. To do so, a transformer diffusion model is modified to take a combination of depth and orientation control signals. First, the method extracts the scene depth using DepthAnything, and computes the occlusion on the desired edited region, determining the location of the 3D object. The object orientation is described with three parameters (azimuth, elevation, in-plane rotation), and added to the object reference condition. To train the transformer, the paper proposes to rely on 3D generative AI data, render the assets in images and generate captions, and then rely on rendering engines' simulations and generative AI placement to synthesize realistic scenes. Final 3D information (depth of generated scene, 2D masks, and positions) is obtained by piping ad-hoc foundational models. Experiments are conducted on 1600 test samples in total, and evaluated using AI-based metrics (e.g., DINO and CLIP similarity) as well as a human user study, surpassing the competitors.

**Strengths:**

- The paper provides a careful combination of different foundational models to generate data, showing impressive results in terms of consistency.
- The paper is well presented and clear
- The data, benchmark, and method could be useful for several downstream tasks and of interest for general ICLR audience

**Weaknesses:**

- The paper relies on an intricate combination of different tools, and not all the details are reported. I believe replicating the results could be particularly challenging, and code release is not mentioned in the paper. The method also requires a non-negligible computational effort for training, as it uses 8xA100 GPUs

- The papers rely on a test set that also uses generated background, DINO features, and other foundational models as judges to assess the consistency of the provided generation. However, since the method itself heavily relies on generative AI, I would argue that an evaluation using more grounded geometrical data would be informative. For example, relying on synthetic rendering or lightstage captured objects could provide an exact ground truth on how the edited image and the rotated object should look, which would also be a nice qualitative result for the reader to inspect the result of the method. On a similar note, it would be interesting to use the method to perform 3D reconstruction of image objects by generating controlled multi-views of it, and then compare the obtained result with the GT Objaverse asset.

**Questions:**

- Will the code be released?
- How long does it take for the training in wall-clock time?
- Could you provide further details on the training, e.g., what are the losses used for? How are the different stages balanced?
- Is the material of the assets considered in the training set creation through Blender? How does the method perform for objects with high reflectance/specularity?

---

> ### Author Response · Authors · 2025-11-21
> **Response to Reviewer XPg4**
>
> ---
> ## W1&Q1: Tool Details, and Code Release
>
> We understand the concern regarding the complexity of the data curation pipeline. Here are the specific details of the tools used:
>
> - **Step 1 (3D Asset Generation)**: We used the default text-to-3D inference settings of Hunyuan-3D 2.0 to generate assets from category names. Captions were produced using Qwen-2.5-VL-7B with the simple prompt: "*Please simply describe this object.*"
>
> - **Step 2 (Placement)**: For simulated placement, we used internal 3D scenes, and random put 1-4 object to the scene and rendering from random view while ensure all the object can be seen. For generative placement, we employed the default inference settings of UNO and prompted ChatGPT-4o with the task: "*Merge the provided multiple objects into a suitable scene with appropriate spatial relationships, forming proper interaction or occlusion relationships.*"
>
> - **Step 3 (Condition Extraction)**: The integrated Grounding-DINO and Segment Anything pipeline from Grounded-SAM[1] was used (using their default implementation). For Orient Anything and Depth Anything, we utilized their ViT-Large version.
>
> **Code Release:** The release of our code and dataset is currently subject to license and legal approval concerning the various tools and data sources utilized. We are actively working to obtain these necessary approvals and will try our best to open-source this project upon acceptance.
>
> [1]: https://github.com/IDEA-Research/Grounded-Segment-Anything
>
> ---
> ## W2: Evaluation using more grounded geometrical data
>
> We completely agree that evaluating against more grounded geometrical data is crucial and informative. **This is precisely how we designed our core evaluation.**
>
> Our object insertion experiment uses a rendered view of a real 3D asset from Objaverse as the object condition. During testing, we compare the edited image against the true appearance of the 3D asset rendered at the specified conditional pose using the DINO score.
>
> This ground truth rendering at the conditional pose provides a reliable and grounded metric to simultaneously measure both ID preservation and pose following accuracy.
>
> Regarding the suggestion to use our method for 3D reconstruction via controlled multi-views: Our model is focused on object-scene interaction and environmental harmony, not single-object multi-view generation (which is the domain of specialized models like InstantMesh[2]). Therefore, it is expected that our method would not outperform them. We view the integration of multi-view information generated by these specialized models (like InstantMesh[2]) as a promising future direction to enhance appearance consistency.
>
> [2] Xu J, Cheng W, Gao Y, et al. Instantmesh: Efficient 3d mesh generation from a single image with sparse-view large reconstruction models[J]. arXiv preprint arXiv:2404.07191, 2024.
>
> ---
> ## Q2&Q3: Further details on the training
>
> The stage 1 requires 120 hours on 8 A100 GPUs, and the stage 2 requires 48 hours on 8 A100 GPUs.
>
> For learning objective, we directly use the standard Flow Matching loss.
>
>
> ---
> ## Q4: Is the material of the assets considered.
>
> Unfortunately, from the training data prespective, the 3D assets used for building our training data were generated by Hunyuan-3D 2.0, and the resulting assets only contained textures without material.
>
> However, we have performed additional experiments to assess the model's ability to handle lighting effects. We provide object insertion results under more varied and challenging lighting conditions in Figure 12 of the updated manuscript. These results demonstrate that our model can harmoniously blend the object with the background's shadow and lighting effects to a certain extent, suggesting an implicit learning of lighting and material consistency.

---

> > ### Comment · Reviewer_XPg4 · 2025-11-23
> > **Post rebuttal**
> >
> > I thank the author for their clarifications.
> >
> > About code release: I do not think that licensing of other tools should affect the code developed within this project, but by my understanding authors are up to release their implementation. Can the authors confirm that they will release the code, for all the components on which they own the licensing?
> >
> > I see the author point about their evaluation: I would like to clarify that my point was about the definition of the metrics, since by my understand all of them rely on an underlying foundational model (Orient Anything, DINO, DepthAnything, QWEN) to provide a ground truth. Is my understanding correct or am I still missing something?
> >
> > I appreciate the further results, while living beings in Figure 11 looks suboptimal and with less consistency than other rigid results in the paper.

---

> > > ### Author Response · Authors · 2025-11-25
> > > **Further Response to Reviewer XPg4**
> > >
> > > Thank you for your prompt and insightful follow-up questions.
> > >
> > > **Code Release**: We confirm that we will release the code for all components of the project where we hold the necessary licensing rights.
> > >
> > > **Evaluation Metrics and Foundational Models**: Yes, your understanding is correct. Our evaluation methodology follows previous object-level editing works (e.g., Anydoor [1]). We rely on foundational models for comparing generated objects against the ground truth.
> > >
> > > **Living Beings in Figure 11**: Our method has not been trained on real human data (the 3D assets used often exhibit a more cartoonish style). The results shown in Figure 11 serve primarily to demonstrate the zero-shot generalization capability of our method to real-world human subjects. Improving consistency and fidelity for living beings is a crucial future direction for this work.
> > >
> > > [1] Chen X, Huang L, Liu Y, et al. Anydoor: Zero-shot object-level image customization. CVPR 2024.

---

### Official Review · Reviewer_9QNt · 2025-10-26

**Soundness:** 3
**Presentation:** 3
**Contribution:** 3
**Rating:** 6
**Confidence:** 4

**Summary:**

This paper presents a method that enable object insertion with 3D control.
The main contribution from my point of view is the dataset curation pipeline and the ability to achieve better result of 3D control.

**Strengths:**

- The dataset generation process is reasonable and serve the purpose of the proposed method well.
- The proposed method is straightforward and easy to understand.
- The demonstrated results seem better than other existing methods.

**Weaknesses:**

- The need of additional training is a bit complicated compared to recent training-free method.
- The computational overhead compared to original base model (e.g., FLUX) is not discussed properly.
- The composited objects sometimes have weird projections or appearance and did not match the scene well, e.g., the airplane in Fig. 5. Regarding that case, I feel the composited results generated by GPT-4o is more convincing.

**Questions:**

- I think overall, composition is a difficult task with a long history in computer graphics and vision. For inserting an object, while the pose is vital, the appearance is also important. Have the authors considered how to better improve the composited appearances? For example, by inputting additional conditions? I think this is not critical but worth discussing in the paper.
- I am curious about how different datasets affect the results? In the paper, there are both simulated placement and generative placement datasets. But it is not clear shown and discussed why both datasets are needed.

---

> ### Author Response · Authors · 2025-11-21
> **Response to Reviewer 9QNt**
>
> ---
> ## W1: Complicated compared to recent training-free method.
> We agree that training-based and training-free approaches each have distinct trade-offs. We argue that our training-based method offers significant advantages in usability and performance:
>
> Our approach provides **more precise pose control** and utilizes a **significantly simpler single diffusion process**. In contrast, prior training-free methods, such as Diffusion Handlers, require a complex and user-unfriendly pipeline. This typically involves intricate steps like point cloud projection&manipulation, diffusion inversion, feature reusing and so on. Furthermore, these complicated steps often fail to guarantee reliable identity preservation or accurate pose following.
>
> Therefore, we believe that the additional training is a worthwhile investment for the substantial benefits it provides to the user in terms of a simpler interaction model, and superior final results.
>
>
> ---
> ## W2: Computational overhead compared to FLUX.
>
> We have measured the practical runtime speed and memory footprint of our method. The efficiency remains largely consistent with the original FLUX model.
>
> For generating an image on an L40 GPU with 25 sampling steps and CFG enabled, **our model takes 33 seconds and utilizes a peak GPU memory of 36 GB**. In comparison, the original FLUX model generating an image from text with **the same settings requires 29 seconds and 25 GB of peak GPU memory**. The minor increase in resource usage is expected and reflects the added complexity for conditions.
>
> ---
> ## W3: The composited objects sometimes did not match well with scene.
>
> We acknowledge that a conflict sometimes arises between object identity preservation and harmonious scene integration when the object's style significantly differs from the scene.
>
> For the specific airplane case in Figure 5, while GPT-4o's result appears more harmonious, it often fails to maintain the consistency of the background or the specified scene. Nano Banana, while better at preserving the background and blending, struggles significantly to understand and follow the target 3D pose.
>
> We think that a stronger foundational model, such as Qwen-Image[1], may help mitigate this limitation. The balance between Object ID preservation, Scene Harmonization, and Pose Following is a critical challenge. Exploring solutions that excel across all three dimensions will be an interesting direction for our future work.
>
> [1] Wu C, Li J, Zhou J, et al. Qwen-image technical report[J]. arXiv preprint arXiv:2508.02324, 2025.
>
> ---
> ## Q1: How to better improve the composited appearances.
>
> We strongly agree with the idea of enhancing composite appearances through additional conditions. We have already explored this in Section 4.3 ("Impact of textual condition"), where we demonstrated that providing textual descriptions of the object can effectively improve its appearance quality.
>
> Beyond this, a promising future direction involves leveraging recent advancements in multi-view single-object generation, like InstantMesh[2]. Whether integrating multi-view consistency information can further improve the appearance quality and consistency of the manipulated object across different poses would be an instersting exploration direction.
>
> [2] Xu J, Cheng W, Gao Y, et al. Instantmesh: Efficient 3d mesh generation from a single image with sparse-view large reconstruction models[J]. arXiv preprint arXiv:2404.07191, 2024.
>
> ---
> ## Q2: Effect of simulated and generative datasets
>
> The two datasets provide complementary benefits essential for the training process:
>
> - **Simulated Placement Data**: This data provides perfectly consistent multi-view appearances of the object but is limited to a narrow range of environments and lighting conditions.
>
> - **Generative Placement Data**: This data offers a more realistic fusion effect and greater diversity in scenes and lighting, though object ID preservation is slightly less rigid.
>
> Our empirical observation during training is that the simulated data ensures better object consistency but can limit the model's generalization to diverse scenes. By incorporating the generative data, we significantly enhance the model's generalization ability while maintaining strong object consistency.
>
> Regrettably, due to data licensing constraints and limited time, we are unable to access the full partition of our data to conduct strict ablation studies on the individual contribution of each dataset type at this time.

---

> > ### Comment · Reviewer_9QNt · 2025-11-21
> >
> > thx for the author rebuttal. I think most of my concerns are addressed.
> > One minor point is that I think 36GB v.s. 25GB is not a minor increase imho.
> > Practically it might requires a totally different level of GPUs.
> > But I understand this is sort of acceptable nowaday, I just think it is important to put this in the paper and share this information during code release in the future, so others can understand this before they try this method.

---

> > > ### Author Response · Authors · 2025-11-25
> > > **Further Response to Reviewer 9QNt**
> > >
> > > Thank you for your support. We will update this computational overhead information in the subsequent version of the manuscript.

---

### Official Review · Reviewer_bQWY · 2025-11-01

**Soundness:** 3
**Presentation:** 3
**Contribution:** 3
**Rating:** 4
**Confidence:** 3

**Summary:**

SpatialHand proposes a practical, single-stage diffusion-transformer framework for 3D-aware object insertion and movement in images by decomposing an object’s 6DoF pose into three conditions: (1) 2D location via a geometry-aware masked scene, (2) target depth via a composited depth map, and (3) 3D orientation embedded as latent tokens (azimuth/elevation/in-plane rotation). They add additional tokens to a pretrained FLUX model, using LoRA to adapt to additional conditions. The authors build a large paired training corpus via (a) synthetic 3D assets + renderer and (b) generative placement using subject-driven models, then use visual foundation models to estimate pose for supervision. They train progressively (identity pretrain → novel-view fine-tune → insertion fine-tune) and show quantitative and qualitative gains for both insertion and movement tasks versus several baselines.

**Strengths:**

* The method presented is elegant and solves critical problems for object manipulation: 3D translation and rotation + in-context generation.
* The scalable data generation pipeline using pretrained 3D model generators + traditional rendering is clearly effective for generating varied data

**Weaknesses:**

* Missing evaluation: I believe the authors should evaluate against LooseControl (SIGGRAPH 2024) and Build-a-Scene (ICLR 2025) which also can move/rotate objects in 3D space.
* I believe the DragDiffusion line of work [CVPR 2024], more recently Dragin3D [CVPR 2025] can also rotate objects and should be evaluated.
* This one is less important, but a recent frontier model Reve.ai can also move objects by placing 2D boxes and could be part of the evaluation.
* The method can handle simple objects, but not humans or stylized artwork (according to the results and nature of the training data). This point should be discussed in the limitation.
* Basic shadows of the rendered object look good, but the object tends to stick out/not always harmonize lighting-wise. Could natural lighting harmonization be evaluated? Could results be presented on diverse/colored lighting conditions? The current scenes all have daytime lighting that is similar to the object’s studio lighting.
* Could practical runtime speed/memory footprint be reported?

**Questions:**

See weaknesses

---

> ### Author Response · Authors · 2025-11-21
> **Response to Reviewer bQWY**
>
> ---
> ## W1 & W2: Why not evaluate LooseControl and Build-a-Scene
>
> The primary reason these two methods were not included in our evaluation is a fundamental incompatibility with our task setting.
>
> - **Primary Reason**: LooseControl and Build-a-Scene rely on re-using features generated during the source image's creation process to maintain object identity (ID). This necessitates that **the source images used for editing must be generated by their own pipeline**. In contrast, our 3D movement setting uses real-world images collected from the internet as source inputs. Therefore, these methods cannot be tested or benchmarked in our experimental setup.
>
> - **Secondary Reason**: Both methods use 3D bounding boxes to represent objects pose, focusing mainly on 3D location. This representation inherently lacks the fidelity required for precise rotation control (e.g., a $x$-degree rotation and a $x+180$-degree rotation are often indistinguishable for a box).
>
> ---
> ## W2: Why not evaluate Dragin3D
> Dragin3D was not included because this project is not open-sourced. Given the complexity of its pipeline, we attempted to reproduce the results but failed to successfully implement and run the method for a fair comparison.
>
> ---
> ## W3: Reve.ai
> We attempted to use Reve.ai for 3D object movement and rotation but were unable to find documentation or a clear mechanism for conditioning the movement using provided 2D bounding boxes.
>
> Furthermore, since Reve.ai appears to be a general-purpose image editing model, similar to advanced models like GPT-4o and Nano Banana, we believe that the inclusion of these two most advanced and widely-used baselines already provides strong evidence of our method's superiority and unique contribution in the domain of precise 3D-aware object manipulation.
>
> ---
> ## W4: Evaluation over human and stylized artwork
> Thanks for your advise. We conducted experiments on cases involving humans and stylized artwork and found that our method is capable of generalizing to these scenarios. Visualizations of these results are provided in Figure 11 of the updated manuscript.
>
> ---
> ## W5: Evaluation over more diverse lighting
> We acknowledge the importance of diverse lighting conditions. We included qualitative results showcasing object insertion under more varied and challenging lighting conditions in Figure 12 of the updated manuscript. These results demonstrate our method's ability to harmoniously blend the object with the background's shadow and lighting effects.
>
> ---
> ## W6: Runtime speed and memory
> We measured the practical runtime speed and memory footprint of our method. The efficiency remains largely consistent with the original FLUX model. For generating an image of the same size on an L40 GPU with 25 sampling steps and CFG enabled, **our model takes 33 seconds and utilizes a peak GPU memory of 36 GB**. In comparison, the original FLUX model generating an image from text with **the same settings requires 29 seconds and 25 GB of peak GPU memory**. The minor increase in resource usage is expected and reflects the added complexity for conditions.

---

> > ### Comment · Reviewer_bQWY · 2025-11-21
> >
> > Thanks for the responses and uploading the revised manuscript. R.e. LooseControl and Build-A-Scene - these papers still solve the same problem and should be evaluated, even if it's only qualitative and the source images are different among the methods. For example, it would be helpful to see the why 180-degree rotations fail with theirs but succeed with yours on different images.
> >
> > R.e. Drag-based methods: there are several recent drag-based methods that solve the same problem, at least one or two should be evaluated (ideally quantitative, but at least qualitative if an apples-to-apples comparison can't be achieved). For example, the DragDiffusion repo can invert images and should run without error; "Drag Your Noise: Interactive Point-based Editing via Diffusion Semantic Propagation" [CVPR 2024] is another recent work; and "Good Drag" [ICLR 2025] is even more recent with code.
> >
> > Thanks for reporting the performance and including the additional figures.

---

> > > ### Author Response · Authors · 2025-11-25
> > > **Further Response to Reviewer bQWY**
> > >
> > > Thanks for your prompt feedback. We have performed the requested comparisons and updated the manuscript accordingly.
> > >
> > > ### **Evaluation against LooseControl and Build-A-Scene:**
> > >
> > > **Quantitative Results:**
> > >
> > > We construct 30 3D object movement testing cases for Build-A-Scene and LooseControl, respectively. The comparative results are presented below, demonstrating that our method significantly outperforms both baselines across key metrics:
> > >
> > > |              | CLIP | AbsRel | Acc@30 | Fidelity | Adherence |
> > > | ------------ | ---- | ------ | ------ | -------- | --------- |
> > > | LooseControl | 62.6 | 20.7   | 28.1   | 3.52     | 2.98      |
> > > | SpatialHand  | **72.2** | **19.5**   | **51.5**   | **4.31**     | **4.43**      |
> > >
> > >
> > > |               | CLIP | AbsRel | Acc@30 | Fidelity | Adherence |
> > > | ------------- | ---- | ------ | ------ | -------- | --------- |
> > > | Build-a-Scene | 63.2 | 20.9   | 29.6   | 3.73     | 3.05      |
> > > | SpatialHand   | **71.5** | **19.9**   | **53.0**   | **4.19**     | **4.23**      |
> > >
> > >
> > > **Qualitative Results:**
> > > We have included a visual comparison in Figure 13 of the updated manuscript. From the qualitative results, we observe that the Box Depth Map used by both Build-A-Scene and LooseControl have an inherent ambiguity when representing object rotation (e.g., the depth maps for a $x^\circ$ and $(x + 180)^\circ$ rotation can be identical). This fundamental limitation restricts their ability to accurately express 3D rotation.
> > >
> > >
> > > ### **Evaluation against Drag-based Methods**
> > >
> > > We have updated Figure 14 in the revised manuscript with a visual comparison against representative drag-based methods, specifically DragDiffusion and Drag Your Noise. We find that drag-based approaches tend to prioritize object deformation to follow the movement trajectory, rather than translating the object while preserving its structural integrity, which is crucial for realistic 3D manipulation.

---

### Author Response · Authors · 2025-12-04
**Summary of the Rebuttal Phase**

We sincerely thank the reviewers and the Area Chair for their valuable time and constructive feedback. This summary details the discussion process and our responses, aiming to provide a clear and transparent reference for the revision.


---
### **Reviewer bQWY (Score: 4 $\to$ ?)**

* **Main Concerns:**
    1.  Need for comparison with additional baselines.
    2.  Potential limitations when handling human subjects or stylized artwork.
    3.  Potential limitations under more diverse lighting conditions.
    4.  Detailed reporting of runtime speed and memory consumption.
* **Our Response:**
    1.  We provided comparative experiments against the requested baselines, which further demonstrated the advantage of our method.
    2.  We provided more quantitative results on human subjects and stylized artwork, confirming good generalization capability.
    3.  We provided more quantitative results under diverse lighting, also showing good generalization.
    4.  We provided detailed inference speed and memory analysis.
* **Outcome Summary:** The reviewer appreciated the further results and requested the additional baseline comparisons. **Although the reviewer did not have a chance to reply or update their score after we provided the extended baseline comparisons, we believe that the clear superiority is enough to address their concern.**

---
### **Reviewer 9QNt (Score: 6 $\to$ 6)**

* **Main Concerns:**
    1.  Comparison with training-free methods.
    2.  Detailed reporting of runtime speed and memory consumption.
    3.  Observation that composited objects sometimes did not match the scene well.
    4.  Discussion on how to further improve composited appearances.
    5.  The specific effect of each dataset used.
* **Our Response:**
    1.  We provided an in-depth comparative discussion involving both training-free and training-based methods.
    2.  We provided detailed inference speed and memory analysis.
    3.  We discussed the reasons why composited objects occasionally show poor scene matching and proposed potential solutions.
    4.  We discussed possible strategies for further improving the composited appearances.
    5.  We clarified the specific role and contribution of each dataset.
* **Outcome Summary:** The reviewer confirmed that **"most of the concerns are addressed" and maintained the positive score.**

---
### **Reviewer XPg4 (Score: 8 $\to$ 8)**

* **Main Concerns:**
    1.  Details regarding tool usage and code release plans.
    2.  Evaluation using grounded geometrical data.
    3.  Need for more detailed training information.
    4.  Whether the material properties of the 3D assets are considered.
* **Our Response:**
    1.  We provided more detail on how each tool was utilized and committed to making our best effort to release the code.
    2.  Our evaluation protocol already partially incorporates geometrically grounded data.
    3.  We provided additional training details.
    4.  We explained that, due to limitations in the 3D asset generation model used, asset material properties are currently not considered.
* **Outcome Summary:** The reviewer acknowledged our clarifications and the additional results provided, and **maintained the positive score.**

---

### Meta-Review · Area_Chair_a78A · 2026-01-07

**Summary:**

The reviewers expressed a positive consensus regarding the acceptance of "SpatialHand," a novel framework for generative object insertion with precise 6DoF control. The paper addresses a significant gap in current generative manipulation methods by effectively decomposing pose into 2D location, depth, and 3D orientation. Key strengths identified by the reviewers include the elegant handling of 3D translation and rotation, the scalable data generation pipeline using synthetic assets and subject-driven models, and the method's superior performance over existing baselines. While there were initial concerns regarding the evaluation against specific baselines, computational overhead, and generalization to non-rigid objects, the authors provided a comprehensive rebuttal with additional experiments and clarifications that solidified the paper's contributions.

**Reviewer Concerns:**

Reviewer Concerns addressed by the rebuttal:

Comparison with additional baselines (Reviewer bQWY): The reviewer requested comparisons against "Loose Control," "Build-a-Scene," and drag-based methods (e.g., DragDiffusion). The authors provided both quantitative and qualitative comparisons in the revision, demonstrating that SpatialHand significantly outperforms these baselines in rotation fidelity and prompt adherence.

Computational overhead and efficiency (Reviewer 9QNt, bQWY): Concerns were raised regarding the runtime and memory footprint compared to the base FLUX model. The authors provided detailed statistics.

Generalization to humans and lighting conditions (Reviewer bQWY, XPg4): Reviewers questioned the model's ability to handle human subjects, stylized artwork, and diverse lighting. The authors added qualitative results demonstrating zero-shot generalization capabilities in these scenarios.

**Reviewer Scores:**

I expect Reviewers XPg4 and 9QNt maintained their positive scores (8 and 6, respectively). Reviewer bQWY (initial score 4) will increase to 6.

---

### Decision · Program_Chairs · 2026-01-26

Accept (Poster)